

# Patterns of bird-window collisions inform mitigation on a university campus

Natalia Ocampo-Peñuela[1], R. Scott Winton[1], Charlene J. Wu[1,2], Erika Zambello[1,3], Thomas W. Wittig[1] and Nicolette L. Cagle[1]

[1] Nicholas School of the Environment, Duke University, Durham, NC, United States
[2] Ecology & Environment Inc., Arlington, VA, United States
[3] Tourism Development Department, Okaloosa County, Fort Walton Beach, FL, United States

## ABSTRACT

Bird-window collisions cause an estimated one billion bird deaths annually in the United States. Building characteristics and surrounding habitat affect collision frequency. Given the importance of collisions as an anthropogenic threat to birds, mitigation is essential. Patterned glass and UV-reflective films have been proven to prevent collisions. At Duke University's West campus in Durham, North Carolina, we set out to identify the buildings and building characteristics associated with the highest frequencies of collisions in order to propose a mitigation strategy. We surveyed six buildings, stratified by size, and measured architectural characteristics and surrounding area variables. During 21 consecutive days in spring and fall 2014, and spring 2015, we conducted carcass surveys to document collisions. In addition, we also collected ad hoc collision data year-round and recorded the data using the app iNaturalist. Consistent with previous studies, we found a positive relationship between glass area and collisions. Fitzpatrick, the building with the most window area, caused the most collisions. Schwartz and the Perk, the two small buildings with small window areas, had the lowest collision frequencies. Penn, the only building with bird deterrent pattern, caused just two collisions, despite being almost completely made out of glass. Unlike many research projects, our data collection led to mitigation action. A resolution supported by the student government, including news stories in the local media, resulted in the application of a bird deterrent film to the building with the most collisions: Fitzpatrick. We present our collision data and mitigation result to inspire other researchers and organizations to prevent bird-window collisions.

Corresponding author
Natalia Ocampo-Peñuela,
no19@duke.edu

## INTRODUCTION

### General bird-window collisions

Bird-window collisions are an important source of anthropogenic bird mortality accounting for as many as one billion bird deaths annually in the United States (*Klem, 1990*; *Klem, 2009a*; *Loss et al., 2014*). Among anthropogenic bird fatalities, window collisions are second only to free-ranging cats (*Loss, Marra & Will, 2015*). Birds flying through urban or

rural landscapes fail to recognize windows as barriers and often collide with them due to glass transparency or reflectivity (*Klem, 1989*). Window collisions are an additional threat for birds that already face natural dangers like predation, disease, starvation, inclement weather, and the cost of long distance migration (*Klem, 2014*). Although it is uncertain whether window collisions are a major cause of the declining trends in some North American bird populations (*Arnold & Zink, 2011*; *DeSante, Kaschube & Saracco, 2015*), mortality due to collisions accounts for an annual loss of 2–9% of the total estimated North American bird population (*Loss et al., 2014*).

## Effects of buildings and surrounding area on collisions

All buildings do not pose an equal threat to birds. From previous studies, glass area of a building has been shown to be the most important feature explaining collisions (*Borden et al., 2010*; *Cusa, Jackson & Mesure, 2015*; *Hager et al., 2013*). Building height also plays a role. Low and medium-rise buildings, such as those found on a university campus, account for 44 and 56% of total collisions in the United States, respectively (*Loss et al., 2014*).

The area surrounding a building is also thought to influence the amount of bird-window collisions by attracting birds to adjacent vegetation or available water sources (*Hager & Craig, 2014*; *Klem, 1989*; *Klem, 2014*). This finding may not apply in all contexts; for example, *Borden et al. (2010)* found that the presence of trees near buildings had no effects on collision presence and frequency.

## Species vulnerability to collisions

While many bird species have been documented as window collision victims, differences in habits and behavior cause some to be far more susceptible than others. Studies in North America have found that 90% of collisions occur during spring and fall migration (*Borden et al., 2010*). Passerines that migrate at night, such as warblers and sparrows, collide with windows frequently (*Arnold & Zink, 2011*; *Gelb & Delacretaz, 2006*; *Klem, 1989*) because they must traverse many stepping stones of unfamiliar habitat in transit between breeding and wintering grounds. Among the migrants, forest understory species, accustomed to flying low and through restricted space between trees, such as thrushes of the genus *Catharus*, Wood Thrush (*Hylocichla mustelina*), Ovenbird (*Seiurus aurocapilla*) and hummingbirds, are among the most common collision victims (*Blem & Willis, 1998*; *Klem, 2014*). The disproportionate effect of window-collisions on migratory species is particularly noteworthy given that 50% of North American migrants have declined by at least 50% over the past 50 years (*Robbins et al., 1989*).

## Mitigation opportunities

Given the importance and frequency of window collisions (*Loss, Marra & Will, 2015*), mitigation options have been both gaining popularity and championed by urban conservationists and architects. Moral/ethical implications notwithstanding, the prevention of collision-caused bird deaths is arguably necessary in order to comply with the Migratory Bird Treaty Act of 1918 and the Endangered Species Act of 1973 (*Klem, 2009a*; *Klem & Saenger, 2013*). There is a wide variety of bird deterrent techniques used on windows, including: glass with etched or sandblasted patterns, fritted glass displaying opaque patterns

on the outer surface, and UV-reflective films. This last solution has the most potential for widespread application, but in order for it to be effective it must reflect 20–40% of incipient radiation between 300 and 400 nm (*Klem, 2009b*), and to date this solution has yet to be systematically tested at the building scale. Patterns that divide the clear space of windows have been proven effective at deterring window collisions when placed no more than 10 cm apart (*Klem, 1990*; *Klem, 2009b*).

## Purpose

The purpose of this study was to investigate the patterns of bird-window collisions at Duke University's campus in Durham, North Carolina. We set out to identify the buildings and building characteristics associated with the highest frequencies of bird-window collisions on campus.

Unlike many research projects, this one was carried out with advocacy in mind. A fundamental goal of this study was to generate an evidence-based foundation from which we could advise Duke University on the scope of bird death on campus, and how it might best be mitigated. Here, we present results on the bird-window collision data, and the resulting mitigation action. If similar projects were to be implemented en masse across the thousands of North American campuses, the aggregate conservation benefit for birds would be substantial. In addition to such direct conservation benefits, the data generated would greatly improve uncertain estimates of bird-window collision mortality and understanding of associated landscape and phenological factors involved.

## METHODS

### Study area

The study was conducted at Duke University's West Campus located in Durham, North Carolina, United States (Fig. 1). Construction of the campus started in 1924 and buildings continue to be added to the 200 existing structures. The suburban campus spans 34 km$^2$, 29 km$^2$ of which are forested. West Campus has a predominantly gothic architecture, though newer buildings include elements of modern construction such as large windows for natural light, multiple wings, and as many as four stories. Starting in 2000, Duke University's administration decided that all new buildings and major renovations would be Leadership in Energy and Environment Design (LEED$^{TM}$) certified, with a goal of earning at least LEED$^{TM}$ Silver status for each (*Campus Sustainability Committee, 2015*).

We selected 6 buildings for the study, stratifying by size: Fitzpatrick Center for Interdisciplinary Engineering Medicine and Applied Sciences (Fitzpatrick), French Family Science Center (French), Penn Pavilion (Penn), Schwartz-Butters Athletic Center (Schwartz), The Perk, and Law School extension (Law extension). Small buildings were <2,500 m$^2$ (The Perk, Law extension), medium sized buildings were between 2,500 m$^2$ and 4,500 m$^2$ (Schwartz, Penn), and large buildings were between 25,000 m$^2$ and 32,000 m$^2$ (French, Fitzpatrick). All buildings except Schwartz are LEED$^{TM}$ certified.

### Carcass surveys

We conducted three carcass surveys during peak migration periods in spring and fall 2014, and spring 2015 following methods described by *Hager & Cosentino (2014)*. We surveyed

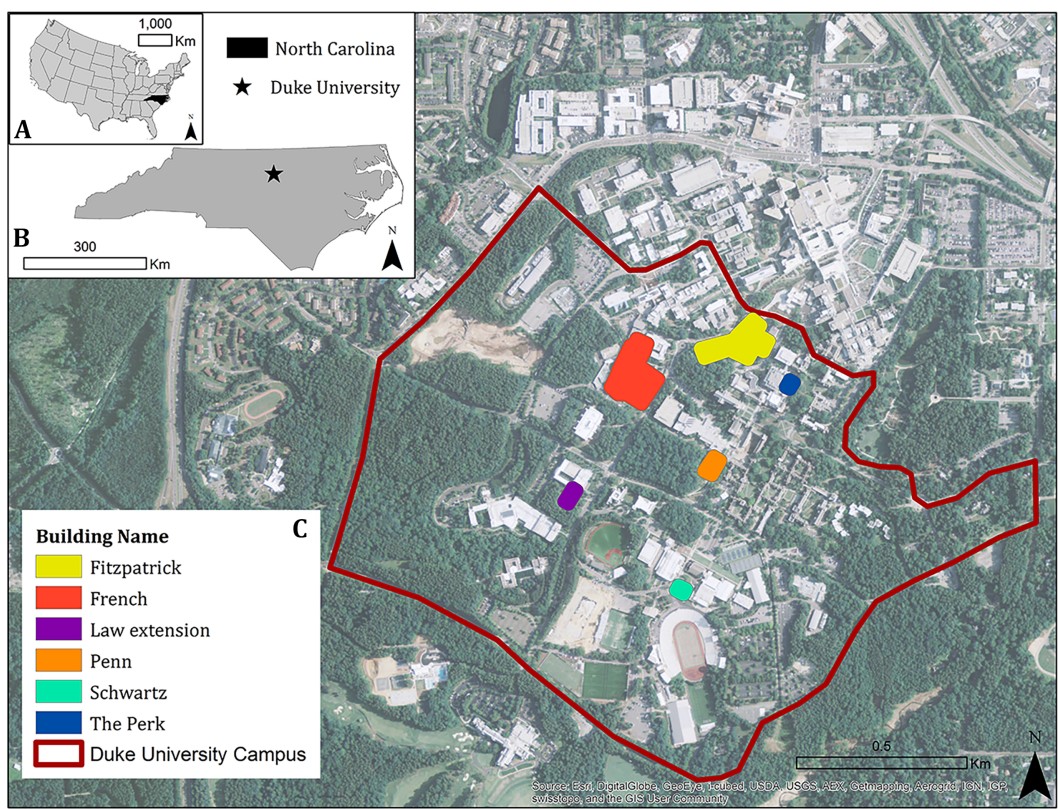

**Figure 1** **Study area.** (A) shows the location of the campus in the United States; (B) within the state of North Carolina. (C) shows Duke University's West campus and the six study buildings. Background image source: *Duke University Facilities Management Department, (2012)*.

the 6 study buildings between 1400 and 1600 h every day for 21 consecutive days. Before the 21-day survey, we picked up all the accumulated carcasses at each building during a clean-up survey, so all buildings started the survey period with zero carcasses. Spring surveys were between April 1st and 21st (clean up March 31st) and the fall survey ran from September 22nd to October 12th (clean-up September 21st). We conducted surveys daily to minimize imperfect detection due to carcass removal by scavengers (*Hager, Cosentino & McKay, 2012*).

During each survey, two observers walked the entire perimeter of each building twice, at a constant speed (1 Km/h), looking for carcasses in a 2-m search swath from the building wall. All carcasses or feather piles were recorded, collected, and deposited in a freezer for identification confirmation (pursuance of Federal Fish and Wildlife Permit MB49165B-0). Some carcasses from the surveys were used for teaching purposes at Duke University, while most of the carcasses were given to the North Carolina Museum of Natural Sciences in Raleigh, NC. We identified all complete carcasses to species, but we left some feather piles unidentified due to uncertainty. Following the data collection protocol proposed by *Hager & Cosentino (2014)*, we recorded data for all surveys, including those in which no birds were found.

Although we only conducted standardized surveys during peak migration times, we collected incidental collision data year-round using the smartphone app and webpage iNaturalist (*Ueda et al., 2015*). Since these data are not standardized, we only used these incidental reports for documenting species richness in bird-window collisions. We only used standardized survey data for all analyses of abundance.

## Buildings and surrounding area

We collected the following data on building traits: floor space (m$^2$), building height (m), total window area (m$^2$), percentage of window area to wall surface (%), LEED$^{TM}$ certification, and presence of a pattern on the glass that could act as bird deterrent.

We used the high resolution (1 m) land cover map for Durham produced by the *US Environmental Protection Agency (2013)* to classify the buildings' surrounding area into three main classes: grass, forest, and impervious. We created land cover thresholds based on percent cover within a 25-m radius. We defined forest and impervious surface as those areas with at least 80% coverage in the 25 m range. Grass had a lower threshold of 25%. With the classified landcover map, we calculated the percentage of area covered by grass, forest, and impervious surfaces within a 50-m buffer around the study buildings.

Because of a small sample size of just six buildings and because two of the sampled buildings dominated the others with respect to total collisions and percent glass area, conventional statistical tests were not appropriate for our building attribute data. Instead, we discuss qualitatively the factors that appear to be associated with collision frequency and drive the outliers.

## Resolution and media coverage

Resolutions are an advocacy tool that allows a community to call attention to an issue and suggest action from the administration. At Duke, the Graduate and Professional Student Council (GPSC) is an important organization for communicating student needs to University administrators. After two seasons of surveys, we wrote a resolution accounting for the documented bird-window collisions on campus to date, and asking Duke University administrators to take action to mitigate bird-window collisions on campus. We presented the resolution to the GPSC General Assembly, which represents more than 8,000 students. The resolution passed unanimously and was sent to all Duke University high level administrators, trustees, and academic deans.

We also agreed to interviews with journalists from the Duke Chronicle, the Raleigh News and Observer, WNCN (local NBC news affiliate), and WRAL (local CBS news affiliate). In addition to the extensive local media coverage, the story of bird-window collisions was the subject of blogs hosted by the Nicholas School of the Environment, the American Birding Association, and Glass Magazine (Data S1).

## RESULTS

The buildings with the most glass area, highest percent glass area, and high surrounding forest cover tended to kill the most birds (Table 1 and Fig. 2). The building with the largest glass area, 57% glass cover and 33% surrounding forest cover, Fitzpatrick, caused 61 of the

86 (71%) collisions detected during standardized surveys (Figs. 2, 3A and 3B). A building with similar amount of glass area but with just 27% of its façade made of glass and little forest cover, French, yielded just 10 collisions (11%), making it the second-most-deadly building of the survey (which it shares with the much smaller Law Extension). The only building in the study with bird deterrent glass, Penn, caused just two window collisions and was the least deadly building in terms of collisions per glass area despite being similar to a glass box (97% glass cover), and in a heavily forested setting (76% surrounded by forest) (Figs. 3C and 3D). Other buildings that caused two or fewer collisions were the two buildings with smallest amount of glass coverage and low surrounding forest cover, Schwartz and The Perk. Schwartz is the only building in the study that is not LEED™ certified.

In addition to the carcasses discovered during our 21-day surveys, we documented 102 incidental collisions throughout the study period across the entire Duke University campus, as well as 33 collisions found during carcass cleanups prior to each survey period. Incidental collisions were most frequently documented during important months for bird migration (April, September, and October) (Fig. 4).

We documented 41 species as collision victims, 31 of which (76%) were migratory. Five species collided with windows five or more times during the standardized carcass surveys: Cedar Waxwing (*Bombycilla cedrorum*) (11), Ovenbird (*Seiurus aurocapilla*) (7), American Goldfinch (*Spinus tristis*) (7), Northern Cardinal (*Cardinalis cardinalis*) (6), and Tufted Titmouse (*Baeolophus bicolor*) (5). Incidental collisions showed a slightly different set of species with the most collisions: Ruby-throated Hummingbird (9), American Goldfinch (8), Yellow-bellied Sapsucker (6), and Hermit Thrush (6) (Table 2).

After collecting these collision data and observing Fitzpatrick's dominant contribution to bird-window collisions, our group, supported by the Graduate and Professional Student Council, led an effort to retrofit Fitzpatrick with bird deterrent patterns. Duke University facilities management department installed a bird deterrent film on several sections of glass façade at Fitzpatrick. Two glass passageways (Fig. 5A) and other windows we identified as dangerous for birds, were retrofitted with a 2.5 cm × 2.5 cm dotted pattern film called Feather Friendly® which is produced by the Canadian-based company *Convenience Group Inc (2015)* (Fig. 5B). Installation was completed in September 2015.

## DISCUSSION

### Building traits, glass, and surrounding area

Our results are consistent with those of previous studies documenting a positive relationship between glass area and window collisions (*Borden et al., 2010*; *Hager et al., 2013*). Buildings on Duke University's campus with more glass tended to cause more bird-window collisions. Fitzpatrick, the building with the most window area, caused the most collisions. Schwartz and the Perk, the two small buildings with small window areas, had the lowest collision frequencies.

The main exception to the correlation between glass area and collision frequency was at Penn, the only building with fritted glass incorporated into the façade. Fritted glass is a

**Table 1  Building traits, surrounding area characteristics and collisions results for six buildings at Duke University's West campus.** Percentage impervious, grass, and forest are based on a 50 m buffer around the building. Days with collisions and total collisions are based on collisions detected during 63 days of standardized surveys in the fall and spring of 2014 and spring of 2015.

| Building name | Building traits | | | | Surrounding area | | | | Collision results | | | |
|---|---|---|---|---|---|---|---|---|---|---|---|---|
| | Floorspace (m²) | Glass area (m²) | Glass cover (%) | LEED™ | Imperv. surface (%) | Grass (%) | Forest (%) | Distance to forest patch (m) | Clean-up survey | Days with collisions | Collisions/ 100 m² glass | Total collisions |
| Fitzpatrick | 30,860 | 1,883 | 57 | Silver | 20 | 47 | 33 | 34 | 19 | 25 | 3.24 | 61 |
| French | 27,282 | 1,716 | 27 | Silver | 60 | 39 | 1 | 102 | 2 | 8 | 0.58 | 10 |
| Schwartz | 4,040 | 148 | 12 | – | 95 | 5 | 0 | 166 | 0 | 2 | 1.35 | 2 |
| Penn[a] | 2,322 | 437 | 98 | Silver | 18 | 6 | 76 | 0 | 0 | 1 | 0.46 | 2 |
| Law extension | 604 | 199 | 56 | Green | 41 | 21 | 39 | 0 | 3 | 2 | 5.03 | 10 |
| The Perk | 416 | 42 | 18 | Green | 74 | 13 | 14 | 218 | 0 | 1 | 2.38 | 1 |

**Notes.**

LEED™, Leadership in Energy and Environmental Design Certification.

[a]Building with pattern on glass.

Ocampo-Peñuela et al. (2016), *PeerJ*, DOI 10.7717/peerj.1652

**Table 2  List of species observed as window collision victims at Duke University's West campus during 2014 and 2015.** Migratory status from *Cornell Lab of Ornithology (2015)*, complemented with local observations.

| Family | Common name | Scientific name | Migrant | # Incid coll. | 2014 | | | | 2015 | | Surv. total |
|---|---|---|---|---|---|---|---|---|---|---|---|
| | | | | | Pre-surv. spring | Surv. spring | Pre-surv. fall | Surv. fall | Pre-surv. spring | Surv. spring | |
| Columbidae | Mourning Dove | *Zenaida macroura* | Resident[a] | 1 | | 1 | | | | | 1 |
| Trochilidae | Ruby-throated Hummingbird | *Archilochus colubris* | Migrant | 9 | | | | 1 | | 2 | 3 |
| Picidae | Downy Woodpecker | *Picoides pubescens* | Resident | 1 | | | | | | | 0 |
| Picidae | Northern Flicker | *Colaptes auratus* | Resident[a] | 1 | | | | | 1 | | 0 |
| Picidae | Yellow-bellied Sapsucker | *Sphyrapicus varius* | Migrant | 6 | | | | 1 | | | 1 |
| Picidae | Red-bellied Woodpecker | *Melanerpes carolinus* | Resident | 3 | | | | | | | 0 |
| Vireonidae | Red-eyed Vireo | *Vireo olivaceus* | Migrant | 2 | | | | 3 | | | 3 |
| Paridae | Tufted Titmouse | *Baeolophus bicolor* | Resident | 1 | 1 | 5 | | | | | 5 |
| Sittidae | White-breasted Nuthatch | *Sitta carolinensis* | Resident | | | 1 | | | | | 1 |
| Troglodytidae | Carolina Wren | *Thryothorus ludovicianus* | Resident | 1 | | | | | | | 0 |
| Regulidae | Golden-crowned Kinglet | *Regulus satrapa* | Migrant | 3 | | | | | | | 0 |
| Regulidae | Ruby-crowned Kinglet | *Regulus calendula* | Migrant | | | 1 | | | | 1 | 2 |
| Turdidae | American Robin | *Turdus migratorius* | Migrant[b] | 1 | 1 | 1 | | | 2 | 2 | 3 |
| Turdidae | Veery | *Catharus fuscescens* | Migrant | | | | 1 | | | | 0 |
| Turdidae | Gray-cheeked Thrush | *Catharus minimus* | Migrant | 1 | | | | 1 | | | 1 |
| Turdidae | Hermit Thrush | *Catharus guttatus* | Migrant | 6 | | | | | 1 | | 0 |
| Turdidae | Wood Thrush | *Hylocichla mustelina* | Migrant | 1 | | | 1 | 3 | | | 3 |
| Turdidae | Swainson's Thrush | *Catharus ustulatus* | Migrant | | | | 1 | | | | 0 |
| Mimidae | Brown Thrasher | *Toxostoma rufum* | Resident[a] | | 1 | | | 2 | | | 2 |
| Mimidae | Northern Mockingbird | *Mimus polyglottos* | Resident | | | | | 2 | | | 2 |
| Mimidae | Gray Catbird | *Dumetella carolinensis* | Migrant | 4 | | | 2 | 3 | | | 3 |
| Bombycillidae | Cedar Waxwing | *Bombycilla cedorum* | Migrant | 2 | 1 | 11 | | | | | 11 |
| Parulidae | American Redstart | *Setophaga ruticilla* | Migrant | 2 | | | | 1 | | 2 | 3 |
| Parulidae | Black-throated Blue Warbler | *Dendroica caerulescens* | Migrant | | | | | 1 | | | 1 |
| Parulidae | Black-throated Green Warbler | *Dendroica virens* | Migrant | | | | | 1 | | | 1 |
| Parulidae | Cape May Warbler | *Dendroica tigrina* | Migrant | 2 | | | | | | | 0 |
| Parulidae | Chestnut-sided Warbler | *Dendroica pensylvanica* | Migrant | 1 | | | | | | | 0 |
| Parulidae | Common Yellowthroat | *Geothlypis trichas* | Migrant | 4 | | | 1 | | | | 0 |
| Parulidae | Ovenbird | *Seiurus aurocapilla* | Migrant | 1 | | | 1 | 2 | | 4 | 6 |
| Parulidae | Yellow-rumped Warbler | *Dendroica coronata* | Migrant | 4 | | 1 | | | | | 1 |
| Emberizidae | White-throated Sparrow | *Zonotrichia albicollis* | Migrant | 2 | | 1 | | | | | 1 |

Ocampo-Peñuela et al. (2016), *PeerJ*, DOI 10.7717/peerj.1652

| Family | Common name | Scientific name | Migrant | # Incid coll. | 2014 | | | | 2015 | | Surv. total |
|---|---|---|---|---|---|---|---|---|---|---|---|
| | | | | | Pre-surv. spring | Surv. spring | Pre-surv. fall | Surv. fall | Pre-surv. spring | Surv. spring | |
| Emberizidae | Eastern Towhee | *Pipilo erythrophthalmus* | Resident[a] | 3 | | | | | | | 0 |
| Emberizidae | Song Sparrow | *Melospiza melodia* | Resident[a] | 4 | | | | | | 1 | 1 |
| Emberizidae | Swamp Sparrow | *Melospiza georgiana* | Migrant | 3 | | | | | | 1 | 1 |
| Emberizidae | Dark-eyed Junco | *Junco hyemalis* | Migrant | 3 | | | | | | 1 | 1 |
| Emberizidae | Fox Sparrow | *Passerella iliaca* | Migrant | 1 | | | | | | | 0 |
| Cardinalidae | Indigo Bunting | *Passerina cyanea* | Migrant | 1 | | | | | | | 0 |
| Cardinalidae | Northern Cardinal | *Cardinalis cardinalis* | Resident | 2 | | 2 | | 2 | 1 | 2 | 6 |
| Cardinalidae | Rose-breasted Grosbeak | *Pheucticus ludovicianus* | Migrant | 1 | | | 1 | | | | 0 |
| Cardinalidae | Scarlet Tanager | *Piranga olivacea* | Migrant | 1 | | | | | | | 0 |
| Fringillidae | American Goldfinch | *Carduelis tristis* | Migrant [b] | 8 | | | | 6 | 1 | 1 | 7 |
| | Unidentified | Unidentified | | 12 | 2 | 7 | 1 | 6 | 2 | 3 | 16 |
| | | **Total** | 31 | 98 | 6 | 31 | 9 | 35 | 8 | 20 | 86 |

**Notes.**

[a] Resident populations on Duke University campus may be augmented by migrants from more northerly latitudes, so it is impossible to determine whether residents and/or migrants of these species are colliding with windows.

[b] Populations are short-distance migrants but some individuals may be local residents so it is impossible to determine whether residents and/or migrants of these species are colliding with windows.

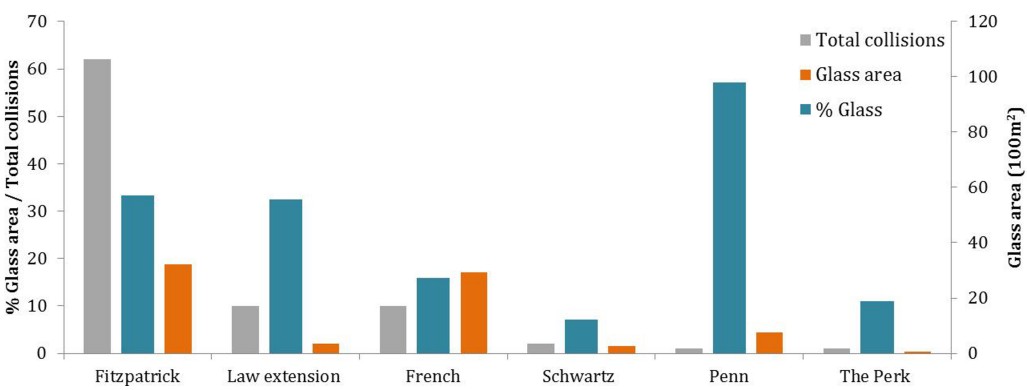

**Figure 2  Glass metrics and bird-window collisions detected during 3 seasons of 21-day surveys of six buildings at Duke University's West campus in Durham, NC.** Penn is the only building in the study with fritted glass known to deter birds.

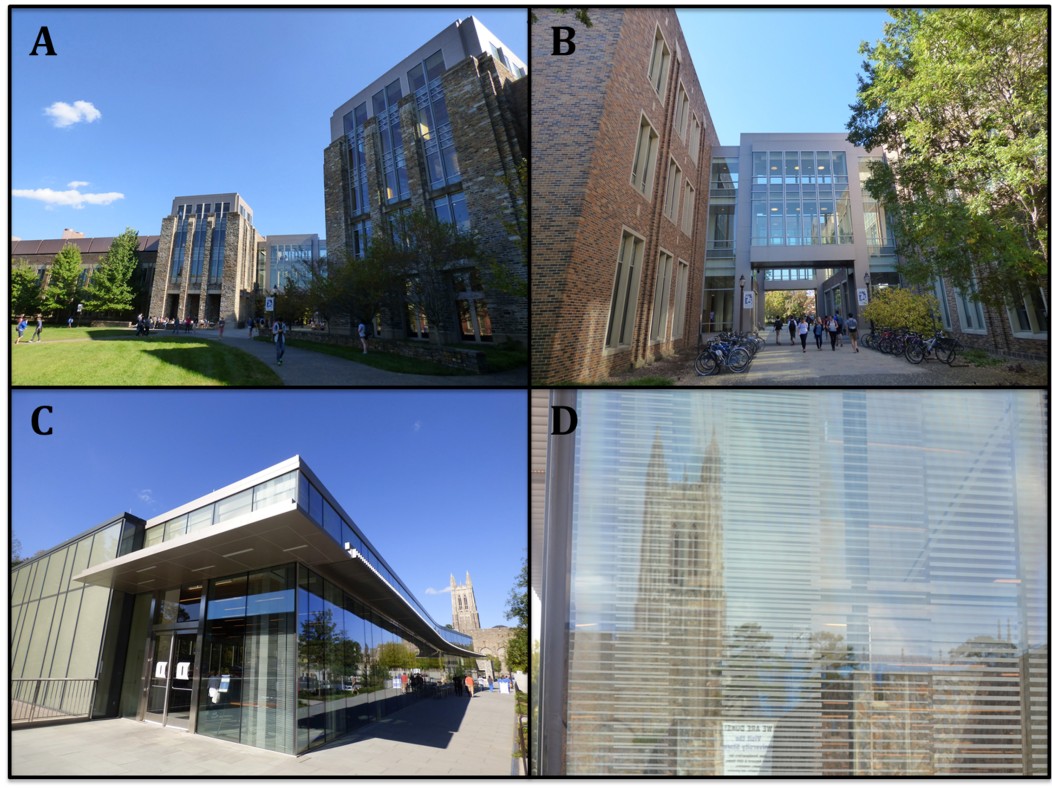

**Figure 3**  (A–B), Fitzpatrick, the buildings with the highest bird-window collision frequency at Duke University. (C–D) Penn, the only building with bird deterrence patterns at Duke University.

feature known to deter bird collisions (*Klem, 1990*). Vertical frit lines cover approximately 30% of Penn's windows (Fig. 3D), which likely helps birds recognize the glass as a barrier mitigating collision incidence.

In addition to glass area, the habitat cover of areas surrounding buildings is also thought to have an effect on the collision susceptibility (*Hager et al., 2013*). We found

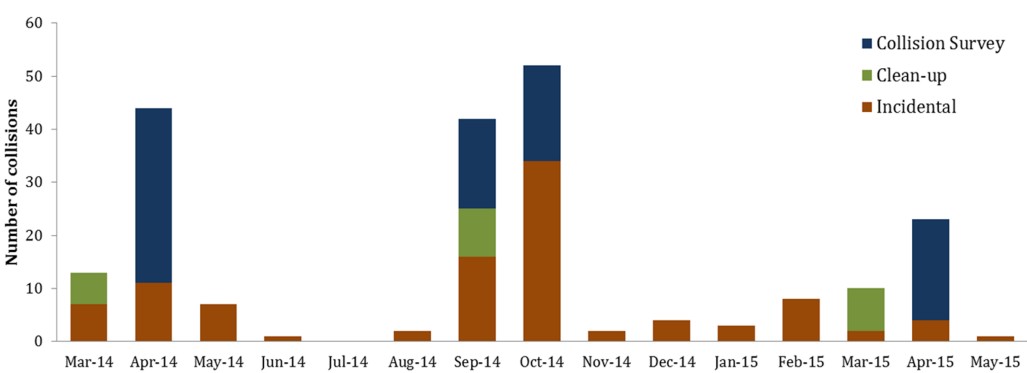

**Figure 4** Seasonal distribution of bird-window collisions binned by month at Duke University's West campus in Durham, NC.

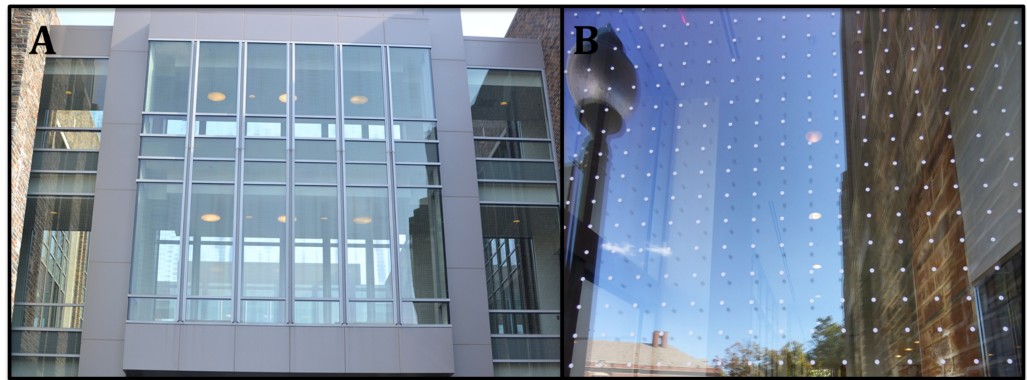

**Figure 5** Bird deterrence dotted patterns on windows of Fitzpatrick building at Duke University. (A) Glass passageways. (B) Close up of dotted pattern. Photos: Casey Collins.

some anecdotal evidence that surrounding area may be interacting with the glass effects we observed at Duke University's campus. For example, Schwartz and the Perk not only have small glass area, but are also surrounded by a high proportion of impervious cover and relatively removed from wooded green spaces, which may have further reduced their susceptibility to collisions. In contrast, Law Extension has a relatively high percentage (39%) of surrounding forest, which may have contributed to a high rate of collisions per unit glass area. If surrounding forest is an important risk factor for bird-window collisions, it makes the relative scarcity of collisions detected at Penn particularly compelling. Not only is the façade of Penn nearly completely made of glass, but the building is partially surrounded by old growth (100+ year-old) forest, which may further indicate the effectiveness of glass fritting in this case.

While the deadliest building, Fitzpatrick, has a moderate amount of surrounding forest cover (33%), we attribute the high total number of collisions it caused to two second-story transparent glass passageways that connect wings of the building (Fig. 5A). While we did not specifically record collision victims from beneath glass passageways, we began to notice that they were a likely site for finding carcasses as we conducted surveys. This observation is consistent with other studies that have implicated glass tunnels as architectural features

associated with high incidence of window collisions (*Agudelo-Álvarez, Moreno-Velasquez & Ocampo-Peñuela, 2010*; *Klem, 1989*).

We noticed a predominance of glass in buildings that are LEED$^{TM}$ certified, which could make these "green" buildings especially deadly to birds. Both Fitzpatrick and Penn are certified at the Silver level and have significant amounts of glass (Table 2). Although LEED$^{TM}$ certified buildings have the potential to be more dangerous for birds (due to high glass area), solutions to prevent collisions could be incorporated as part of the certification process. American Bird Conservancy has already advocated for a LEED credit to prevent window collisions (*US Green Buildings Council, 2011*) but we encourage more research on the impact of the certification on collisions, and recommend this issue be weighted more heavily in the certification scheme.

## Seasonality

From our year-round campus-wide incidental collision data, we observed a trend of higher bird-window collisions during spring and fall migration, especially during September and October (Fig. 3). On a campus in Ohio, where similar research took place, 90% of deaths by collisions also occurred during migration (*Borden et al., 2010*). We confirm that standardized surveys during peak migration, as proposed by *Hager & Cosentino (2014)*, is an efficient way of gathering collision data. We recommend augmentation of their survey method by adding a spring survey to the protocol because it improves chances to detect some species that may be missed in the fall due to differences in migratory behaviors in the two seasons.

## Species vulnerability

Although collisions occur year-round and can impact a wide range of bird species, migratory species appear to be particularly vulnerable (*Blem & Willis, 1998*; *Borden et al., 2010*; *Klem, 2009a*). Our data supports the idea that migratory birds are especially susceptible to window-collision mortality, as we found that 76% of the species recorded during carcass surveys were migratory and an additional 9% were partially migratory. One migratory species, Cedar Waxwing, was involved in more collisions than any other species, accounting for 17% of the total collisions detected during surveys. Cedar Waxwing is a gregarious species during migration (*Sibley, 2003*) and when collisions occurred, we found several individuals simultaneously. This species may be particularly vulnerable to collisions because of the consumption of fermented berries that can cause ethanol toxicosis affecting the bird's flight and sense of orientation (*Fitzgerald, Sullivan & Everson, 1990*). The second most common collision victim on Duke University campus, the Ovenbird, is listed by many studies of bird-window collisions as one of the most frequently encountered species (*Blem & Willis, 1998*; *Borden et al., 2010*; *Cusa, Jackson & Mesure, 2015*; *Hager et al., 2008*). The Ovenbird is an understory specialist, a guild which has been identified as highly vulnerable to collisions (*Blem & Willis, 1998*).

The non-migratory species we most frequently observed as collision victims were Northern Cardinal and Tufted Titmouse. Other studies have noted the pattern that migrants collide most frequently during migration, whereas permanent residents are at risk of collision year-round (*Blem & Willis, 1998*).

### Retrofitting of Fitzpatrick

The combination of sound scientific data, media coverage, and a resolution supported by representatives of more than 8,000 students (approximately half of the total student body), led Duke University to take action to mitigate bird deaths on campus (Fig. 5). Scientific data allowed us to identify problem buildings and prioritize windows for retrofitting treatment. Media coverage helped communicate a local problem to a wider audience, and contributed to convincing the university to take action. The GPSC resolution helped us reach high level administrators, which may have otherwise been insulated from this issue. An additional research project we participated in allowed us to put Duke University's collision data in context. A collaboration led by Hager and Cosentino aimed to evaluate the drivers of bird-window collisions in North America at 40 university campuses. Duke University was the campus with the highest collision frequency (S Hager & B Cosentino, 2015, unpublished data), which contributed to our call to action.

Conservation biology is described as a 'crisis science' (*Soulé, 1985*), but all too often biological research ends for the scientist at the publication stage and crises remain unsolved. Here, we have presented a rare example of conservation research that progressed almost immediately from data collection to mitigation. We caution that action did not happen serendipitously, but rather we engaged with decision makers and communicated with the media. This required effort beyond the scope of the standard research life cycle, but we encourage other researchers, particularly those in conservation biology, to follow our example and engage media, peers, and decision-makers to resolve the crises being studied.

### Recommendations

Bird-window collision studies have looked at patterns of presence and frequency of collisions as a snap-shot, but research that compares time of collision, different seasons, years, or even decades are still lacking. We recommend collision surveys that collect data over migratory and non-migratory seasons, and for consecutive years. Another factor that has been overlooked in the analysis of collisions patterns is the weather. From studies about migration, we know that bird movements can be affected by the weather (*Richardson, 1990*), yet we still ignore how it can affect the frequency of bird-window collisions.

Monitoring the effectiveness of bird deterrent materials is fundamental to management of buildings and their effect on wildlife. Additionally, testing these materials at the building scale and evaluating the effectiveness of UV-reflective materials is still needed. When available, placing camera traps near windows might help with documenting the timing of collisions, as well as mapping exact locations of collision events to better inform prevention.

## ACKNOWLEDGEMENTS

We thank the many Duke University graduate and undergraduate students who assisted with data collection for this project, especially the Wildlife Surveys class at the Nicholas School of the Environment. We also thank the many volunteers around campus who have collected data for the collision project since 2013, especially Anna Wilson. John Gerwin

of the North Carolina Museum of Natural Sciences assisted by receiving all bird carcasses we collected. We thank the 2014–2015 General Assembly of the Graduate and Professional Student Council for unanimously passing our bird-window collisions resolution. Duke University Facilities and Management Department deserves credit for its willingness to understand our study and take action on behalf of campus birds by retrofitting Fitzpatrick.

### Funding
The authors received no funding for this work.

### Competing Interests
The authors declare there are no competing interests.

### Author Contributions
- Natalia Ocampo-Peñuela conceived and designed the experiments, performed the experiments, analyzed the data, contributed reagents/materials/analysis tools, prepared figures and/or tables, reviewed drafts of the paper.
- R. Scott Winton conceived and designed the experiments, performed the experiments, analyzed the data, wrote the paper, prepared figures and/or tables, reviewed drafts of the paper.
- Charlene J. Wu, Erika Zambello and Thomas W. Wittig performed the experiments, analyzed the data, wrote the paper, prepared figures and/or tables, reviewed drafts of the paper.
- Nicolette L. Cagle conceived and designed the experiments, performed the experiments, analyzed the data, contributed reagents/materials/analysis tools, wrote the paper, prepared figures and/or tables, reviewed drafts of the paper.

### Animal Ethics
The following information was supplied relating to ethical approvals (i.e., approving body and any reference numbers):
Federal Fish and Wildlife Permit MB49165B-0.

### Data Availability
The raw data was supplied as Data S2.

### Supplemental Information
Supplemental information for this article can be found online at http://dx.doi.org/10.7717/peerj.1652#supplemental-information.

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
