# Peer review of "Patterns of bird-window collisions inform mitigation on a university campus"

_PeerJ, doi:10.7717/peerj.1652_

## Round 0.1 · original submission · Major Revisions

Please consider all of the reviewer suggestions in your revised manuscript

·

Basic reporting

Overall basic reporting is well written: clear, concise, and accurate.

Experimental design

Experimental design and consideration of limitations accounted for (explained and justified).

Validity of the findings

Findings are valid: supporting and cross-validating several previously published studies. Strength of manuscript is in bold and courageous objective of advocacy for a scientific-based study having relevance to local geography (campus) and planet-wide reach at similar sites worldwide.

Additional comments

The following are suggested corrections/recommended modifications addressing inappropriate references (errors) or omissions.

1. Throughout text, recommend deleting “Jr “in “Klem Jr” citations.

2. Line 27. Klem Jr 1989 citation is in error and should be Klem Jr 1990 (see 1 above); recommend adding Klem Jr 2009a or preferably Klem 2009a (see 1 above).

3. Line 66. Add Klem Jr 2009a or preferably Klem 2009a (see 1 above).

4. Line 222. Klem Jr 1990 should be Klem Jr 2009b or preferably Klem 2009b (see 1 above).

5. Lines 364-367, in Literature Cited section. Unless missed, found no Klem Jr 2009a citation in text, but recommend placement in Lines 27, 66 (see 2, 3 above).

Reviewer 2 ·

Basic reporting

No comments.

Experimental design

No Comments.

Validity of the findings

The study is adequately conducted. An improved approach might be to compare the retrofitted Fitzpatrick building after bird deterrent film was applied to windows.

Additional comments

Review: Patterns of bird-window collisions inform mitigation on a university campus
Overall
The authors conducted a one-year study (April 2014-April 2015) to examine the frequency and composition of window kills at six buildings on the Duke University campus. They sampled daily for 3 weeks in April and September and on an ad hoc basis over the rest of the period. The authors found that the more glass per unit area in a building, the more bird deaths there were. Windows with bird deterrent patterns successfully repelled birds from striking windows. The authors used their findings to convince administrators at Duke University to retrofit windows with bird deterrent film.

Comments
This is a well conducted study with meaningful results. The authors have convincingly provided data to support their hypothesis that increased glass surface area results in higher proportions of collisions. Laudably, the authors used their preliminary findings to effect conservation change at the campus.

The manuscript does need some cleaning up. Here are some suggested line edits
3, Remove “presence and” and simply write “…affect collision frequency”
4, Replace “human-caused” with “anthropogenic”
7, Remove “on campus”
10, Suggest you reword sentence as “In addition to focal data collection, we also collected ad hoc collision data year-round and recorded the data using the app iNaturalist”.
12, Presumably the authors mean “glass area” when the write simply “glass”.
16, Better style to write “supported” than “backed up”
17, Suggest “including” rather than “plus”
18, Better to avoid the overdramatic “deadliest” and simply reword as “…to the building with the most collisions, Fitzpatrick”.
18, “result” should be plural “results”
26, Better to replace “human-caused” with “anthropogenic”
27, Do not include “Jr” in the Klem reference. This should be corrected throughout.
28, Better to replace “human caused” with “anthropogenic”. In this case, human-caused was missing the hyphen. Likewise, “free-ranging” also needs a hyphen.
30, Better to use “collide with” as opposed to “collide against”
41, Reword “have the highest cumulative number of collisions” with simply “account for”
52, Citations are not presented in chronological order. 1989 should come before 2006, which should come before 2011.
55-56, Odd order of presentation (species, genus, species, taxonomic group).
61, Remove “importance”, suggest inserting “and frequency “ so the sentence reads “Given the importance and frequency of window collisions”.
61-62, Ambiguous reference to “its apparent ubiquity”.
83, Suggest changing “site is” to “was conducted at”.
98, Difficult to distinguish the French color red from The Perk color pink on the map.
107-109, Unclear what PeerJ’s preferred style for dates is but usually most journals prefer “01 April” over “April 1st”.
112, Suggest “constant rate” rather than “constant speed”. The authors should also indicate what this rate was.
112, “2-m” should be hyphenated
113-115, The commentary on how specimens were used seems unnecessary. Suggest deleting this sentence.
117-118, Suggest “we recorded data for all surveys” rather than “we filled data forms for all surveys”.
122, Remove “at Duke University”. The authors have already indicated where the study took place on line 83.
124, Colloquial use of the term “random”. The authors mean “ad hoc”.
124-125, This sentence is redundant with what has already been said 112-113. Combine sentences.
132, “50-m” should be hyphenated.
138, 140 Unclear what criterion the authors used to determine that outliers were present.
149, Insert serial comma after “trustees” for consistency.
173, Not sure if the authors are planning to include the table with their plot. If so, it should be created using standard table style – not as part of a figure.
176, The buildings appear to be ordered by total collisions. As such, the total collisions bar should appear first in the series. Likewise, stylistically it would be better to have glass area appear last so it is closest to the right axis.
179, I think a word got omitted from the figure caption. Perhaps the authors meant to write “study with fritted”
186-187, Typically one would list months in chronological order: April, September, October.
193, Scientific name for Cedar Waxwing should be given at first mention, not down at line 165. Ditto for Ovenbird (here instead of down at line 271).
194, American Goldfinch, Northern Cardinal, Tufted Titmouse scientific names should be included here.
200, Table 2. Suggest changing column titles “Clean-up Spring” and “Clean-up Fall” to “Pre-survey Spring” or “Pre-survey Fall”. Table itself needs to be formatted correctly. No gridlines, for example. Many of the species listed under Migrant* do not breed in Durham. But, the footnote suggests any birds in the Migrant column are both resident at Duke and are migrants. In the case of Chestnut-sided Warblers, for example, there is no ambiguity. NO Chestnut-sided warblers are found in Durham during the breeding season, nor would any be considered “resident populations”. I think the authors should separate out the migrant column. Rather than putting the asterisk on the column header, stick the asterisk with the x on the row. So, for example, American Robins would be listed as “migrant*” but Gray-cheeked Thrush would be listed simply as “migrant”.
207, Use metric.
208, typo in Convenience.
211, Suggest rewording as “Bird deterrence dotted patterns on windows of Fitzpatrick…” rather than “in Fitzpatrick”.
225, Suggest “effect” rather than “impact” given the use of “collision” in the sentence.
227-231, The authors switch tenses: “…Perk not only have…” vs “…Law Extension had a…”
230, Suggest “of surrounding forest” rather than the awkward sounding “of forested surrounding”.
231, Suggest “per unit glass area”
232, “then” is unnecessary when following a comma.
239, Suggest “record” rather than the colloquial “keep track of”. Suggest deleting both “collected” and “the” and inserting a comma following “passageways,”.
242, Probably no need to use a hyphen in “window collisions”.
243, Suggest “predominance” rather than “dominance”. Incorrect use of a restrictive clause – a comma should precede “, which”
244, Suggest rewording as “Both Fitzpatrick and Penn are certified at the Silver level and have significant amounts of glass (Table 2).
246-247, Suggest rewording as “solutions to prevent collisions” rather than “collision prevention solutions”; suggest removing “also”; suggest “as part of the certification process” rather than “to these “green” structures”.
254, Normally one refers to “on campus”, as opposed to “in campus”.
258, Unclear why some species may be missed? Is this a sampling error, observer bias, different migration paths?
260-261, Suggest rewording the end of the sentence after the comma with “migratory species appear to be particularly vulnerable”.
265, Waxwing was already introduced earlier in the manuscript.
280, Replace the colloquial “backed up” with “supported”.
283, Prioritize what? The sentence ends prematurely.
284, Poor style. Reword “put pressure on the school’s reputation”.
285, Insert comma before “, which” to avoid incorrect use of a restrictive clause.
288, Suggest changing “in” to “at”.
289, Another incorrect use of a restrictive clause. A comma should precede “, which”.
294, Avoid the colloquialism “reached out”.
296, Insert serial comma after “peers,” to match style used throughout.
304, A reference is needed after the statement that bird movements are affected by weather.
305-306, Suggest removing the last sentence of this paragraph – it is unnecessary.
307-308, Suggest rewording the sentence as follows “…fundamental to management of buildings and their effect on wildlife”.
310, Change “document” to “documenting”.
313, Suggest changing “masters” to “graduate” to match “undergraduate”.
Literature Cited
Check reference titles – some use sentence style capitalization while others capitalize each word.
Arnold and Zink reference incomplete.
Campus Sustainability reference is missing closing parenthesis.
Convenience and Friendly spelled incorrectly.
Cusa et al missing volume.
DeSante reference incomplete.
ESRI reference incomplete.
Fitzgerald reference needs journal name to be capitalized, volume number missing.
Loss et al. 2015 incomplete.
Sibley incomplete.
US Green Buildings reference incomplete.
Supplementary data – the data format for the first 4 rows is different (day/month) than the rest of the table (month/day).

Reviewer 3 ·

Basic reporting

Audience may understand better the association between the directions of bird migration and spatial distribution of bird-window collisions if author provide the map of bird-window collisions by different species.

Experimental design

Because of limited data samples, authors did not use statistical models to support their hypothesis. Most of these findings are sample data description. I do not think the paper which only reported the sample data description can be accepted by the academic journals. More sophisticated statistical methods (such as non-parametric statistics or Bayesian models) could be used to test the hypothesis for small data samples.

Validity of the findings

Only relying on data description, limited data samples and observation periods, I doubt the findings of the study can support their hypothesis.

Additional comments

Regarding Fig.4, I noticed that the number of collisions in spring 2015 is significantly less than the number in spring 2014. Do authors provide any explanations or discussions on this issue?

---

## Round 0.2 · Minor Revisions

Please consider the final additional suggestion of Reviewer 3, who suggests that you elaborate on the significance in your Introduction.

Reviewer 3 ·

Basic reporting

the paper is written clearly.

Experimental design

No comments.

Validity of the findings

No comments

Additional comments

Authors responded that there is no any specific hypothesis to test and confirmatory analysis and inferential statistics were also not used in this paper. The goal of the paper is to describe the distribution of the samples collected from the single campus. Authors mentioned "A fundamental goal of this study was to generate an evidence-based foundation from which we could advise Duke University..." The results truly met the goal validly. I think the paper meets the PeerJ criteria. But i still would like suggest authors elaborate the scientific significance of the study (only reporting samples) in the INTRODUCTION. For example, authors may need to explain why this kind of descriptive study or reporting samples is important for global scientific community.

---

## Round 0.3 · accepted · Accept

Congratulations for your work.